# Sweetened beverages and risk of frailty among older women in the Nurses' Health Study: A cohort study

**Ellen A. Struijk**[1,2]*, **Fernando Rodríguez-Artalejo**[1,2,3], **Teresa T. Fung**[4,5], **Walter C. Willett**[5,6], **Frank B. Hu**[5,6], **Esther Lopez-Garcia**[1,2,3]

**1** Department of Preventive Medicine and Public Health, School of Medicine, Universidad Autónoma de Madrid-IdiPaz, Madrid, Spain, **2** CIBERESP (CIBER of Epidemiology and Public Health), Madrid, Spain, **3** IMDEA/Food Institute, CEI UAM+CSIC, Madrid, Spain, **4** Department of Nutrition, Simmons University, Boston, Massachusetts, United States of America, **5** Department of Nutrition, Harvard T.H. Chan School of Public Health, Boston, Massachusetts, United States of America, **6** Channing Division of Network Medicine, Department of Medicine, Brigham & Women's Hospital and Harvard Medical School, Boston, Massachusetts, United States of America

* ellen.struijk@uam.es

## Abstract

### Background

Consumption of sugar-sweetened beverages (SSBs) has been consistently associated with a higher risk of obesity, type 2 diabetes, cardiovascular disease, and premature mortality, whereas evidence for artificially sweetened beverages (ASBs) and fruit juices on health is less solid. The aim of this study was to evaluate the consumption of SSBs, ASBs, and fruit juices in association with frailty risk among older women.

### Methods and findings

We analyzed data from 71,935 women aged ≥60 (average baseline age was 63) participating in the Nurses' Health Study (NHS), an ongoing cohort study initiated in 1976 among female registered nurses in the United States. Consumption of beverages was derived from 6 repeated food frequency questionnaires (FFQs) administered between 1990 and 2010. Frailty was defined as having at least 3 of the following 5 criteria from the FRAIL scale: fatigue, poor strength, reduced aerobic capacity, having ≥5 chronic illnesses, and weight loss ≥5%. The occurrence of frailty was assessed every 4 years from 1992 to 2014. During 22 years of follow-up, we identified 11,559 incident cases of frailty. Consumption of SSBs was associated with higher risk of frailty after adjustment for diet quality, body mass index (BMI), smoking status, and medication use, specifically, the relative risks (RRs) and 95% confidence interval (95% CI) for ≥2 serving/day versus no SSB consumption was 1.32 (1.10, 1.57); $p$-value <0.001. ASBs were also associated with frailty [RR ≥2 serving/day versus no consumption: 1.28 (1.17, 1.39); $p$-value <0.001]. Orange juice was associated with lower risk of frailty [RR ≥1 serving/day versus no consumption: 0.82 (0.76, 0.87); $p$-value <0.001], whereas other juices were associated with a slightly higher risk [RR ≥1 serving/day versus no consumption: 1.15 (1.03, 1.28); $p$-value <0.001]. A limitation of this study is that,

**Data Availability Statement:** Information including the procedures to obtain and access data from the Nurses' Health Studies is described at https://www.

nurseshealthstudy.org/researchers (contact email: nhsaccess@channing.harvard.edu).

**Funding:** This work was supported by grants from the Instituto de Salud Carlos III, State Secretary of R+D+I of Spain and FEDER/FSE (FIS 16/609, 16/1512, 19/319) (FRA and ELG); the European Union (JPI A Healthy Diet for a Healthy Life, SALAMANDER project) (FRA); and the Nurses´ health study is supported by grant UM1 CA186107 from National Institutes of Health (http://www.nih.gov/). The funders had no role in study design, data collection and analysis, decision to publish, or preparation of the manuscript.

**Competing interests:** The authors have declared that no competing interests exist.

**Abbreviations:** 95% CI, 95% confidence interval; AHEI, Alternate Healthy Eating Index; ASB, artificially sweetened beverage; BMI, body mass inde; FFQ, food frequency questionnaire; MET, metabolic equivalent task; NHANES, National Health and Nutrition Examination Survey; NHS, Nurses' Health Study; RR, relative risk; SSB, sugar-sweetened beverage; STROBE, Strengthening the Reporting of Observational Studies in Epidemiology.

due to self-reporting of diet and frailty, certain misclassification bias cannot be ruled out; also, some residual confounding may persist.

## Conclusions

In this study, we observed that consumption of SSBs and ASBs was associated with a higher risk of frailty. However, orange juice intake showed an inverse association with frailty. These results need to be confirmed in further studies using other frailty definitions.

## Author summary

### Why was this study done?

- Frailty is a geriatric syndrome with multiple causes and contributors, which is manifested by fatigue, diminished strength, and reduced physical functioning and leads to a higher risk of dependency and death.

- Due to the aging of the population, an increasing number of people are at risk of developing frailty. Therefore, identifying determinants of frailty is important to support evidence-based preventive interventions.

- So far, there is little information on whether consumption of sugar-sweetened beverages (SSBs), artificially sweetened beverages (ASBs), and fruit juices influences the risk of frailty.

### What did the researchers do and find?

- We studied the association of consumption of SSBs, ASBs, and fruit juices with the risk of frailty among of 71,935 older women participating in the Nurses' Health Study (NHS).

- During 22 years of follow-up, a higher consumption of SSBs and ASBs was associated with a higher risk of frailty, whereas higher orange juice consumption was associated with a lower risk. These associations were independent of lifestyle, medication use, and the quality of the rest of the diet.

### What do these findings mean?

- This study suggests that habitual SSBs drinking increases the risk of frailty in older women. Due to the high SSBs intake and its many adverse health effects, possibly including frailty, older adults should be advised to limit SSBs consumption.

- It is unclear why ASBs were associated with frailty risk. Further research should assess this association and its mechanisms.

## Introduction

Frailty is a geriatric syndrome characterized by a progressive decline in physiological systems and functional reserves that leads to a high risk of falls, disability, hospitalization, and death [1,2]. This syndrome involves functional limitation, unintentional loss of weight, malnutrition and, in many cases, partly results from the synergistic effect of several diseases [3]. Due to the aging of the population, an increasing number of people is expected to suffer this condition in the coming decades [4]. Thus, it is important to identify the determinants of frailty to ensure that older adults not only live longer but also maintain healthier lives as they age.

Research on dietary factors associated with frailty is still limited. Some specific components of the human diet, including fruit, proteins, and micronutrients, are thought to decrease frailty risk when consumed in adequate amounts [5–7]. Moreover, dietary patterns with overall good quality have been associated with lower risk of frailty [8,9]. However, the effects of specific food components in low-quality diets are not clear.

Added sugar intake constitutes a significant portion of the US diet, providing an average percentage of daily energy intake of 13.6% among older adults [10]. In addition, 65% of older adults exceed the 10% maximum recommended by the World Health Organization and the 2015 Dietary Guidelines Advisory Committee [10–12]. The largest contribution to added sugar intake in the American diet is from liquid sources including sugar-sweetened beverages (SSBs) (37.1%) and fruit drinks (8.9%) [10]. Consumption of added sugar from these liquid sources may not suppress the intake of solid foods in subsequent meals and thus, energy balance can be altered toward higher total energy intake and weight gain [13]. On the other hand, consumption of foods and beverages high in added sugar could displace nutrient-rich components of the diet, increasing the risk of malnutrition in the older population. These mechanisms, together with the adverse effect of sugar on inflammation, glucose tolerance, and lipid metabolism [14,15], may partially link sugary beverages to adverse health outcomes including diabetes, heart disease, premature mortality [16,17] and, possibly, frailty. SSBs are often replaced by artificially sweetened beverages (ASBs). About 23.1% of women aged 60 and older in the US population consumed ASBs on a given day during 2009 to 2010 [18]. The effects of these beverages on health are not well established.

We hypothesized that higher consumption of sugary beverages is associated with higher risk of frailty in older adults. Therefore, we investigated the association of SSBs, ASBs, and fruit juices with the risk of frailty in a large population of older women from the Nurses' Health Study (NHS).

## Methods

### Ethics statement

The Harvard T.H. Chan School of Public Health and the Brigham and Women's Hospital Human Subjects Committee Review Board approved the protocol for the study, and participants provided written informed consent. There was no formal prospectively written protocol for the current study. All analyses described below were decided a priori, except for the additional adjustment for physical activity and baseline morbidity, the combined analysis of SSBs and ASBs, and the sensitivity analysis defining the weight loss component as 10% weight reduction, which were suggested by the reviewers.

### Study design and participants

The NHS was established in 1976 with the enrollment of 121,700 female nurses aged 30 to 55 years at inception [19]. Participants completed biennial mailed questionnaires to update

information on medical history and lifestyle. The follow-up rate was approximately 90% at each follow-up cycle.

## Dietary assessment

Dietary intake was assessed using a validated food frequency questionnaire (FFQ) administered every 4 years as described in detail elsewhere [20]. For the current study, we used the FFQs prior to frailty assessment in 1990, 1994, 1998, 2000, 2006, and 2010. In each questionnaire, participants were asked how often on average during the previous year they had consumed the foods specified. A standard portion size and 9 possible responses for the frequency of consumption, ranging from "never, or less than once per month" to "6 or more times per day" were given for each food item. The consumptions of the following beverages were summed as SSBs: caffeinated and non-caffeinated colas (e.g., Coke, Pepsi, and other colas with sugar), other carbonated beverages with sugar (e.g., 7 Up), and noncarbonated sweetened beverages (e.g., Hawaiian Punch, lemonade, and other noncarbonated fruit drinks). In addition, ASBs consisted of caffeinated, caffeine-free, and noncarbonated low-calorie or diet beverages. Fruit juices included orange juice, apple juice or cider, grapefruit juice, prune juice, and non-specified fruit juices. To best represent long-term diet during follow-up and to account for changes in food consumption, we used the cumulative average consumption of these beverages from all available dietary questionnaires from baseline through frailty onset or the end of follow-up [21]. We stopped updating diet information when a participant reported a diagnosis of diabetes during follow-up to exclude changes in sugary beverage consumption as a consequence of this endpoint. Correlation coefficients between FFQs and multiple dietary records for SSBs were 0.84 for colas, 0.36 for non-cola carbonated soft drinks, 0.56 for noncarbonated sweetened beverages, and 0.84 for fruit juice [22].

Nutrient intakes were calculated by multiplying the consumption of each food recorded with the FFQs by its nutrient content, using the US Department of Agriculture database and complemented with information from the manufacturers. Total energy and nutrient intakes were calculated by summing the derived intakes from all foods. Previous research showed that, compared with multiple dietary records, 24-h dietary recalls, and biomarkers of diet, the FFQ provides sufficient information to detect important associations with disease [23,24]. A modified Alternate Healthy Eating Index (AHEI) score was used as an indicator of overall diet quality. This score was calculated based on 10 foods and nutrients that are predictive of chronic disease risk, including fruit, vegetables, nuts and legumes, red and processed meat, whole grains, alcohol, sodium, trans fat, long-chain omega-3, and other polyunsaturated fats, and excluding the item for SSBs consumption [25]. A higher score in the AHEI denotes better diet quality (range 0 to 10).

## Frailty assessment

We used the FRAIL scale [26] that includes 5 self-reported frailty criteria: fatigue, poor strength (reduced resistance), reduced aerobic capacity, having several chronic illnesses, and significant weight loss during the previous year. In 1992, 1996, 2000, 2004, 2008, and 2012, the participants completed the Medical Outcomes Study Short-Form (SF-36), a 36-item questionnaire with 8 health dimensions, including physical and mental components [27]. From the SF-36, we assessed the first 3 frailty criteria with the following questions: (1) for fatigue: "Did you have a lot of energy?," with response options "some of the time" or "none of the time" or with the question "I could not get going," with response options "moderate amount" or "all of the time"; (2) for poor strength: "In a normal day, is your health a limitation to walk up 1 flight of stairs?," with responses "yes" or "a lot"; and (3) for reduced aerobic capacity: "In a normal day,

is your health a limitation to walk several blocks or several miles?," with response options "yes" or "a lot." In addition, the illnesses criterion was assessed from the question "In the last 2 years, have you had any of these physician-diagnosed illnesses?." We considered that this criterion was met when participants reported ≥5 of the following diseases: cancer, hypertension, type 2 diabetes, angina, myocardial infarction, stroke, congestive heart failure, asthma, chronic obstructive lung disease, arthritis, Parkinson disease, kidney disease, and depression. Finally, because weight of the participants was available only biannually, the weight loss criterion was defined as a ≥5% decrease in the weight reported in a 2-year period before the assessment of frailty. At the end of each follow-up cycle, incident frailty was defined as having ≥3 criteria in the FRAIL scale. The recovery rate of frailty was 14%, 6%, and 1% after respectively 4, 8, and 12 years of follow-up, which indicates that frailty is a stable outcome. Despite the absence of performance-based measures, the FRAIL scale has been shown to be correlated with the Fried scale (r = 0.617, $p$ <0.001) [28], the most widely used scale for frailty assessment, which includes both self-reported and performance-based measures, among older adults in care settings.

## Ascertainment of mortality

Deaths were reported by the next of kin, the postal system, or ascertained through the National Death Index. Follow-up for mortality was more than 98% complete [29]. We obtained copies of death certificates and medical records to determine causes of death (classified according to the International Classification of Diseases, Ninth Revision). Death records were reviewed and coded by physicians.

**Medical history, anthropometric data, and lifestyle factors.** From the 1992 questionnaire, we collected information on age, weight, smoking status, and medication use. This information has been updated on each of the subsequent biennial questionnaires. To calculate body mass index (BMI), we used information on height measured in 1976, when the cohort was initiated; BMI was calculated as weight in kilograms divided by the square of height in meters. Discretionary physical activity was reported as the average time spent per week during the preceding year in specific activities (e.g., walking outdoors, jogging, and bicycling). The time spent in each activity was multiplied by its typical energy expenditure, expressed in metabolic equivalent tasks (METs), and then summed overall activities. Detailed information on the validity and reproducibility of self-reported weight and physical activity has been published elsewhere [30,31].

**Statistical analysis.** For this analysis, we included women aged ≥60 years at baseline with complete information on the exposure and outcome variables. Women younger than 60 years at baseline in 1992 entered the study when they turned 60 during follow-up. Women with an unreasonably high (>3,500 kcal/d) or low (<500 kcal/d) caloric intake were excluded from follow-up, as well as women identified as frail at analytical baseline, leaving a total of 71,935 women for the analysis. The association between sweetened beverages and frailty occurrence was examined up to 2014.

Participants were classified into 6 groups according to sweetened beverage consumption: never or almost never (reference), 1 to 3 servings per month, 1 serving per week, 2 to 6 servings per week, 1 to 2 servings per day, and 2 or more servings per day. Since orange juice consumption represents 65% of total juices reported, a separate analysis for this beverage and a combination of all other juices was performed. We used cause-specific proportional hazards models [32] to calculate relative risks (RRs), approximated by hazard ratios, and their 95% confidence interval (95% CI) for the association between each category of sweetened beverage consumption and frailty, adjusting for potential confounders updated at each 4-year time period.

Person-years were calculated from baseline until the occurrence of frailty, death, or the end of the study period (1 June 2014), whichever came first. The Andersen–Gill (counting process) data structure was used to handle time-varying covariates and left truncation [33]. We stratified the analysis jointly by age in years at start of follow-up and calendar year of each questionnaire cycle. Multivariable models were adjusted for BMI at baseline (<25.0, 25.0 to 29.9, and $\geq$30.0 kg/m$^2$), baseline physical activity (in quintiles of METs-h/wk), smoking status (never, past, and current with 1 to 14, 15 to 24, and $\geq$25 cigarettes/day), energy intake (quintiles of kcal/d), alcohol intake (0, 1.0 to 4.9, 5.0 to 14.9, or $\geq$15.0 g/d), and current medication use (yes/no) including postmenopausal hormone therapy, aspirin, diuretics, beta blockers, calcium channel blockers, angiotensin converting enzyme inhibitors, other antihypertensive medication, statins, and other cholesterol-lowering drugs, insulin, and oral hypoglycemic medication. Medication use was included in the model to address the fact that persons with risk factors for chronic disease are possibly at greater risk of developing frailty, although some over adjustment might exist. Similarly, the inclusion of BMI might account for some over adjustment because weight loss is part of the frailty outcome. In addition, we adjusted for diet quality using the AHEI (quartiles of the score). Because it might cause some over adjustment, baseline diseases (heart disease, diabetes mellitus, and cancer) have been added to a separate model. All models were mutually adjusted for the other types of beverages to obtain estimates for a beverage independent of the other beverages consumed. Physical activity is closely related to the outcome; therefore, analyses have been repeated excluding this variable from the model. Linear trends were evaluated using the Wald test on a continuous variable using median intakes of each category of beverage consumption. The risk of frailty associated with 1 serving/d increment in beverage consumption was also calculated. Moreover, the association between sweetened beverage consumption and each criterion of the FRAIL scale was examined separately.

Stratified analyses were done by age (<70 versus $\geq$70 y), BMI (<25 versus $\geq$25 kg/m$^2$), physical activity (below versus above the median), and the AHEI (below versus above the median). Interaction was evaluated using the Wald test on cross-product terms based on beverage intake (continuous variable) and the stratification variable.

In sensitivity analysis, only the most recent measurement of beverage consumption was considered in relation to frailty. Also, analyses among women with 0 frailty criteria at baseline were performed to understand whether the effect of beverage consumption on frailty may differ depending on the baseline frailty status. In addition, an analysis including dietary exposure before baseline measured in 1980, 1984, and 1986 in association with the risk of frailty and 6-, 8-, and 12-year lagged analyses were performed. Although the FRAIL scale includes having several diseases as 1 of the frailty criteria, additional analyses were performed excluding women with diabetes, heart disease, or cancer at baseline or those who developed these diseases during the follow-up to assess the independence of the studied associations from main chronic diseases. To evaluate the FRAIL scale including only those with a more severe weight loss, we have performed analysis in which we defined weight loss as a 10% weight reduction in 2 years.

All statistical tests were 2-sided with a $p$-value <0.05 and performed using SAS software version 9.4 for UNIX (SAS Institute, Cary, North Carolina, US). This manuscript follows the Strengthening the Reporting of Observational Studies in Epidemiology (STROBE) recommendations (S1 STROBE Checklist) [34].

## Results

In Table 1, the age-standardized baseline characteristics of the study participants by categories of sweetened beverages are presented. Compared to women in the lowest category of

**Table 1. Characteristics of women at study entry[a], by categories of sweetened beverages consumption, in the NHS.**

| | SSBs | | | ASBs | | | Total fruit juices | | |
|---|---|---|---|---|---|---|---|---|---|
| | Never or almost never | 1/wk | ≥2/d | Never or almost never | 1/wk | ≥2/d | Never or almost never | 1/wk | ≥2/d |
| Participants, n | 28,981 | 10,231 | 1,074 | 22,312 | 7,542 | 5,329 | 7,290 | 8,703 | 3,143 |
| Mean age, y | 62.9 (2.4) | 62.6 (2.2) | 62.4 (2.1) | 63.0 (2.5) | 62.7 (2.3) | 62.2 (2.0) | 63.0 (2.4) | 62.5 (2.2) | 62.9 (2.5) |
| BMI, kg/m² | 26.7 (5.2) | 26.4 (5.0) | 27.2 (5.9) | 25.2 (4.8) | 26.5 (4.8) | 28.9 (5.9) | 26.6 (5.2) | 27.0 (5.3) | 25.7 (5.0) |
| Current smoker, % | 10 | 10 | 21 | 15 | 7 | 11 | 13 | 12 | 11 |
| Discretionary physical activity, METs-h/wk | 20.7 (24.2) | 19.8 (23.9) | 17.0 (22.6) | 19.6 (23.2) | 20.5 (22.8) | 17.9 (22.7) | 19.5 (23.8) | 18.5 (21.8) | 23.2 (27.8) |
| Medication use[b] | | | | | | | | | |
| Aspirin, % | 49 | 49 | 48 | 47 | 50 | 50 | 47 | 49 | 50 |
| Postmenopausal hormone therapy, % | 36 | 33 | 26 | 32 | 35 | 31 | 36 | 33 | 33 |
| Diuretics, % | 11 | 11 | 12 | 9 | 11 | 14 | 10 | 11 | 12 |
| β-Blockers, % | 14 | 14 | 14 | 13 | 14 | 16 | 13 | 14 | 15 |
| Calcium channel blockers, % | 11 | 10 | 11 | 9 | 10 | 13 | 11 | 10 | 11 |
| ACE inhibitors, % | 10 | 10 | 12 | 9 | 10 | 12 | 9 | 10 | 11 |
| Other blood pressure medication, % | 9 | 8 | 11 | 8 | 9 | 10 | 9 | 9 | 8 |
| Statins, % | 18 | 18 | 24 | 15 | 18 | 23 | 17 | 19 | 17 |
| Other cholesterol-lowering drugs, % | 4 | 4 | 6 | 3 | 4 | 5 | 4 | 5 | 3 |
| Insulin, % | 3 | 1 | 2 | 1 | 1 | 5 | 3 | 2 | 1 |
| Oral hypoglycemic drugs, % | 4 | 2 | 5 | 2 | 2 | 8 | 4 | 3 | 3 |
| Cancer, % | 6 | 5 | 5 | 6 | 6 | 6 | 6 | 6 | 6 |
| Heart disease, % | 4 | 3 | 4 | 3 | 3 | 5 | 4 | 3 | 4 |
| Diabetes, % | 6 | 2 | 4 | 2 | 3 | 10 | 6 | 4 | 4 |
| Number of frailty criteria, % | | | | | | | | | |
| 0 | 75 | 75 | 66 | 76 | 76 | 66 | 74 | 72 | 76 |
| 1 | 20 | 20 | 26 | 19 | 20 | 26 | 21 | 23 | 19 |
| 2 | 5 | 4 | 8 | 4 | 4 | 8 | 5 | 5 | 5 |
| Dietary intake | | | | | | | | | |
| SSBs, s/d | 0.00 (0.00–0.00) | 0.18 (0.14–0.21) | 2.50 (2.36–2.94) | 0.14 (0.00–0.48) | 0.07 (0.00–0.23) | 0.00 (0.00–0.10) | 0.00 (0.00–0.07) | 0.07 (0.00–0.21) | 0.14 (0.00–0.52) |
| ASBs, s/d | 0.43 (0.05–1.00) | 0.14 (0.00–0.64) | 0.00 (0.00–0.20) | 0.00 (0.00–0.00) | 0.17 (0.14–0.21) | 2.57 (2.42–3.22) | 0.24 (0.00–1.00) | 0.28 (0.00–0.87) | 0.07 (0.00–0.57) |
| Fruit juice, s/d | 0.43 (0.07–1.00) | 0.64 (0.25–1.07) | 0.64 (0.16–1.14) | 0.64 (0.15–1.07) | 0.59 (0.20–1.04) | 0.39 (0.07–0.97) | 0.00 (0.00–0.00) | 0.18 (0.14–0.21) | 2.50 (2.14–2.93) |
| Orange juice, s/d | 0.14 (0.00–0.49) | 0.36 (0.07–0.79) | 0.25 (0.07–0.79) | 0.29 (0.07–0.79) | 0.32 (0.07–0.79) | 0.14 (0.02–0.57) | 0.00 (0.00–0.00) | 0.07 (0.07–0.14) | 1.02 (1.00–2.50) |
| Other fruit juices, s/d | 0.07 (0.00–0.21) | 0.14 (0.06–0.42) | 0.14 (0.02–0.50) | 0.14 (0.00–0.43) | 0.14 (0.02–0.35) | 0.07 (0.00–0.23) | 0.00 (0.00–0.00) | 0.07 (0.04–0.14) | 1.07 (0.33–1.76) |
| Energy intake, kcal/d | 1,576 (1,301–1,886) | 1,771 (1,484–2,101) | 2,229 (1,870–2,567) | 1,714 (1,409–2,069) | 1,691 (1,403–2,024) | 1,726 (1,412–2,085) | 1,479 (1,208–1,790) | 1,596 (1,327–1,925) | 2,086 (1,760–2,440) |
| AHEI score | 52.4 (45.9–58.9) | 50.0 (43.9–56.1) | 43.0 (37.4–49.3) | 49.8 (43.0–56.8) | 51.6 (45.3–57.8) | 47.9 (41.8–54.5) | 51.0 (43.6–57.9) | 49.5 (43.2–56.0) | 51.4 (44.7–57.8) |

(*Continued*)

**Table 1.** (Continued)

| | SSBs | | | ASBs | | | Total fruit juices | | |
|---|---|---|---|---|---|---|---|---|---|
| | Never or almost never | 1/wk | ≥2/d | Never or almost never | 1/wk | ≥2/d | Never or almost never | 1/wk | ≥2/d |
| Alcohol intake, g/d | 1.8 (0.0–8.7) | 1.5 (0.0–6.5) | 0.0 (0.0–2.5) | 1.1 (0.0–6.7) | 1.5 (0.0–6.7) | 0.9 (0.0–6.5) | 0.9 (0.0–5.8) | 1.2 (0.0–6.5) | 0.1 (0.0–6.7) |

ACE, angiotensin converting enzyme; AHEI, Alternate Healthy Eating Index; ASB, artificially sweetened beverage; BMI, body mass index; IQR, interquartile range; MET, metabolic equivalent task; NHS, Nurses' Health Study; s, serving; SD, standard deviation; SSB, sugar-sweetened beverage.

Values are means (SD), dietary intake values are medians (IQR), unless otherwise indicated. Data, except age, were directly standardized to the age distribution of the entire cohort.

[a] Entry was age 60.

[b] One or more times per week.

consumption, those with higher consumption of SSBs or ASBs had higher BMI and were less physically active. By contrast, high fruit juice consumption was associated with lower BMI and more physical activity. Medication use was similar across strata, although the use of insulin and oral hypoglycemic drugs was remarkably high among those women in the highest category of ASBs consumption. Total energy intake increased across the categories of SSBs and fruit juices, whereas the AHEI score and alcohol intake were lower only among those with higher SSB consumption.

During 22 years of follow-up, we identified a total of 11,559 incident frailty cases among the 71,935 women of this study (Table 2). SSB consumption was associated with higher risk of frailty after adjustment for lifestyle factors and medication use. The RRs (95% CI) across categories of increasing consumption were 1.00, 1.00 (0.95, 1.05), 1.09 (1.03, 1.16), 1.11 (1.05, 1.17), 1.33 (1.21, 1.46), and 1.46 [(1.22, 1.74); $p$-value <0.001]. Additional adjustment for diet quality and baseline morbidity somewhat attenuated the association. By contrast, ASB consumption was also associated with higher risk of frailty in fully adjusted models [RRs across categories of increasing consumption 1.00, 0.99 (0.93, 1.05), 1.00 (0.93, 1.06), 1.05 (1.00, 1.11), 1.11 (1.04, 1.19), and 1.28 [(1.17, 1.39); $p$-value <0.001]. Joint analyses showed that higher consumptions of both beverages simultaneously had also a direct association with frailty, in comparison with the lowest consumption of both beverages [RR for highest tertile versus lowest tertile: 1.18 (1.08, 1.29); $p$-value <0.001].

Fruit juices were associated with lower risk of frailty [RRs: 1.00, 0.97 (0.89, 1.06), 0.94 (0.87, 1.03), 0.92 (0.86, 1.00), 0.88 (0.81, 0.95), and 0.91 (0.79, 1.04); $p$-value 0.01]. This inverse association was entirely due to orange juice consumption [≥1 s/d versus never or almost never: 0.82 (0.76, 0.87); $p$-value <0.001], whereas other types of juices showed a slight positive association [≥1/d versus never or almost never: 1.15 (1.03, 1.28); $p$-value <0.001]. Excluding physical activity from the models did not change the results.

We found a significant interaction for SSB and orange juice with age; however, the stratified results do not show large differences in estimates for women aged <70 compared to women aged ≥70. Results did not vary strongly across other subgroups in the stratified analyses (Table 3). Additionally, the associations between sweetened beverages and each frailty criterion are shown in Fig 1. Both SSBs and ASBs were associated with a higher risk of all the individual frailty criteria, whereas orange juice was associated with lower risk of the fatigue, poor strength, and reduced aerobic capacity criteria.

When only the most recent information on beverage consumption before the development of frailty was used, we still observed an increased risk of frailty for higher SSBs and ASBs

**Table 2. RRs (95% CI) of frailty according to categories of sweetened beverages consumption among 71,935 women aged ≥60 y in the NHS.**

| | Never or almost never | 1/mo to 3/mo | 1/wk | 2 to 6/wk | 1-2/d | ≥2/d | P for trend | Per 1 serving/d increase |
|---|---|---|---|---|---|---|---|---|
| **SSBs** | | | | | | | | |
| Participants, n | 28,981 | 14,715 | 10,231 | 13,492 | 3,442 | 1,074 | | |
| Person-year | 372,001 | 244,222 | 160,434 | 210,786 | 42,745 | 10,435 | | |
| Frailty cases, n | 3,926 | 2,604 | 1,890 | 2,461 | 545 | 133 | | |
| Age adjusted | 1.00 | 0.97 (0.93, 1.02) | 1.09 (1.03, 1.16) | 1.17 (1.11, 1.23) | 1.52 (1.39, 1.67) | 1.98 (1.66, 2.36) | <0.001 | 1.32 (1.27, 1.38) |
| Multivariable model[a] | 1.00 | 1.00 (0.95, 1.05) | 1.09 (1.03, 1.16) | 1.11 (1.05, 1.17) | 1.33 (1.21, 1.46) | 1.46 (1.22, 1.74) | <0.001 | 1.18 (1.13, 1.23) |
| Multivariable model[b] | 1.00 | 0.97 (0.93, 1.03) | 1.05 (0.99, 1.11) | 1.04 (0.99, 1.10) | 1.22 (1.11, 1.34) | 1.32 (1.10, 1.57) | <0.001 | 1.12 (1.07, 1.17) |
| Multivariable model[c] | 1.00 | 0.98 (0.94, 1.04) | 1.06 (1.00, 1.12) | 1.05 (1.00, 1.11) | 1.23 (1.12, 1.35) | 1.32 (1.10, 1.57) | <0.001 | 1.12 (1.07, 1.18) |
| **ASBs** | | | | | | | | |
| Participants, n | 22,312 | 7,885 | 7,542 | 19,542 | 9,325 | 5,329 | | |
| Person-year | 324,602 | 145,224 | 117,297 | 292,308 | 109,672 | 51,520 | | |
| Frailty cases, n | 3,292 | 1,614 | 1,297 | 3,334 | 1,294 | 728 | | |
| Age adjusted | 1.00 | 1.04 (0.98, 1.11) | 1.12 (1.05, 1.19) | 1.28 (1.22, 1.35) | 1.63 (1.52, 1.74) | 2.32 (2.14, 2.52) | <0.001 | 1.29 (1.26, 1.31) |
| Multivariable model[a] | 1.00 | 0.98 (0.93, 1.04) | 1.00 (0.93, 1.06) | 1.07 (1.01, 1.12) | 1.15 (1.07, 1.22) | 1.36 (1.25, 1.48) | <0.001 | 1.11 (1.09, 1.14) |
| Multivariable model[b] | 1.00 | 0.99 (0.93, 1.05) | 1.00 (0.94, 1.07) | 1.06 (1.01, 1.11) | 1.12 (1.05, 1.20) | 1.31 (1.20, 1.42) | <0.001 | 1.10 (1.07, 1.12) |
| Multivariable model[c] | 1.00 | 0.99 (0.93, 1.05) | 1.00 (0.93, 1.06) | 1.05 (1.00, 1.11) | 1.11 (1.04, 1.19) | 1.28 (1.17, 1.39) | <0.001 | 1.09 (1.06, 1.12) |
| **Total fruit juices** | | | | | | | | |
| Participants, n | 7,290 | 7,534 | 8,703 | 25,587 | 19,678 | 3,143 | | |
| Person-year | 78,221 | 103,854 | 116,549 | 424,287 | 280,634 | 37,077 | | |
| Frailty cases, n | 840 | 1,216 | 1,356 | 4,995 | 2,840 | 312 | | |
| Age adjusted | 1.00 | 0.96 (0.87, 1.04) | 0.92 (0.84, 1.00) | 0.84 (0.78, 0.90) | 0.78 (0.72, 0.85) | 0.78 (0.68, 0.89) | <0.001 | 0.89 (0.86, 0.92) |
| Multivariable model[a] | 1.00 | 0.97 (0.89, 1.06) | 0.95 (0.87, 1.03) | 0.92 (0.85, 0.99) | 0.88 (0.81, 0.95) | 0.90 (0.79, 1.03) | <0.001 | 0.96 (0.92, 0.99) |
| Multivariable model[b] | 1.00 | 0.96 (0.88, 1.05) | 0.94 (0.86, 1.03) | 0.92 (0.85, 0.99) | 0.88 (0.81, 0.95) | 0.91 (0.80, 1.04) | 0.001 | 0.96 (0.93, 0.99) |
| Multivariable model[c] | 1.00 | 0.97 (0.89, 1.06) | 0.94 (0.87, 1.03) | 0.92 (0.86, 1.00) | 0.88 (0.81, 0.95) | 0.91 (0.79, 1.04) | 0.01 | 0.96 (0.93, 0.99) |
| | Never or almost never | 1/mo to 3/mo | 1/wk | 2 to 6/wk | ≥1/d | | | Per 1 serving/d increase |
| **Orange juice** | | | | | | | | |
| Participants, n | 14,390 | 13,028 | 8,696 | 22,275 | 13,546 | | | |
| Person-year | 174,406 | 169,201 | 132,442 | 391,756 | 172,817 | | | |
| Frailty cases, n | 2,040 | 1,844 | 1,572 | 4,517 | 1,586 | | | |
| Age adjusted | 1.00 | 0.93 (0.87, 0.99) | 0.88 (0.82, 0.94) | 0.81 (0.76, 0.85) | 0.78 (0.73, 0.83) | | <0.001 | 0.83 (0.80, 0.87) |
| Multivariable model[a] | 1.00 | 0.95 (0.89, 1.01) | 0.92 (0.86, 0.98) | 0.87 (0.82, 0.92) | 0.83 (0.77, 0.89) | | <0.001 | 0.89 (0.86, 0.93) |
| Multivariable model[b] | 1.00 | 0.94 (0.88, 1.00) | 0.91 (0.85, 0.98) | 0.87 (0.82, 0.91) | 0.82 (0.76, 0.88) | | <0.001 | 0.89 (0.85, 0.93) |
| Multivariable model[c] | 1.00 | 0.94 (0.89, 1.01) | 0.92 (0.86, 0.98) | 0.87 (0.82, 0.92) | 0.82 (0.76, 0.87) | | <0.001 | 0.89 (0.85, 0.93) |
| **Other juices[d]** | | | | | | | | |

*(Continued)*

**Table 2.** (Continued)

|  | Never or almost never | 1/mo to 3/mo | 1/wk | 2 to 6/wk | 1-2/d | ≥2/d | P for trend | Per 1 serving/d increase |
|---|---|---|---|---|---|---|---|---|
| Participants, n | 21,345 | 15,762 | 12,954 | 17,393 | 4,481 |  |  |  |
| Person-year | 268,995 | 256,046 | 197,037 | 272,042 | 46,502 |  |  |  |
| Frailty cases, n | 2,904 | 2,960 | 2,218 | 3,058 | 419 |  |  |  |
| Age-adjusted | 1.00 | 1.01 (0.96, 1.06) | 0.98 (0.92, 1.03) | 1.01 (0.96, 1.07) | 1.07 (0.96, 1.19) |  | 0.22 | 1.02 (0.96, 1.08) |
| Multivariable model[a] | 1.00 | 1.05 (0.99, 1.10) | 1.02 (0.97, 1.08) | 1.09 (1.03, 1.15) | 1.12 (1.00, 1.24) |  | 0.004 | 1.07 (1.01, 1.14) |
| Multivariable model[b] | 1.00 | 1.05 (1.00, 1.11) | 1.04 (0.98, 1.10) | 1.13 (1.07, 1.19) | 1.16 (1.05, 1.29) |  | <0.001 | 1.11 (1.05, 1.17) |
| Multivariable model[c] | 1.00 | 1.06 (1.00, 1.11) | 1.04 (0.98, 1.10) | 1.13 (1.07, 1.19) | 1.15 (1.03, 1.28) |  | <0.001 | 1.10 (1.04, 1.17) |

[a] Adjusted for age (years), calendar time (4-y intervals), BMI (<25.0, 25.0–29.9, ≥30.0 kg/m$^2$), smoking status (never, past, and current 1–14, 15–24, and ≥25 cigarettes/day), alcohol intake (0, 1.0–4.9, 5.0–14.9, or ≥15.0 g/d), energy intake (quintiles of kcal/d), physical activity (quintiles), and medication use (aspirin, postmenopausal hormone therapy, diuretics, β-blockers, calcium channel blockers, ACE inhibitors, other blood pressure medication, statins and other cholesterol-lowering drugs, insulin, and oral hypoglycemic medication).

[b] Adjusted for variables in model a and additionally adjusted for the AHEI (quartiles).

[c] Adjusted for variables in model b and additionally adjusted for cancer, heart disease, and diabetes (yes/no). All beverages were mutually adjusted for each other.

[d] This group includes apple juice or cider, grapefruit juice, prune juice, and non-specified fruit juices

95% CI, 95% confidence interval; ACE, angiotensin converting enzyme; AHEI, Alternate Healthy Eating Index; ASB, artificially sweetened beverage; BMI, body mass index; NHS, Nurses' Health Study; RR, relative risk; SSB, sugar-sweetened beverage.

consumption. Orange juice remained inversely associated with frailty (S1 Table). In addition, analysis among the women without frailty criteria at baseline showed similar associations (S2 Table), as well as analysis including cumulative diet information including the time period before baseline, and also latency analysis (S3 and S4 Tables). Finally, when excluding women with heart disease, cancer, or diabetes or when using a weight loss criterion defining weight loss as a 10% weight reduction in 2 years, the association between the intake of sweetened beverages and frailty remained similar (S5 and S6 Tables).

## Discussion

In this analysis of a large prospective cohort in the US, we found that habitual consumption of SSBs and ASBs was associated with higher risk of frailty, whereas orange juice was associated with lower risk. The relationships were independent of lifestyle, medication use, and diet quality and remained similar across different subgroups of women.

The association between SSBs and frailty showed a positive association across increasing categories of consumption, especially in the 2 highest categories (above 1 serving a day). Of note is that participants in the NHS had an average SSBs intake of 0.23 (SD 0.41) servings a day, which is lower than the average intake from the nationally representative population of the National Health and Nutrition Examination Survey (NHANES) in the same age category in 2009 to 2010 (0.61 servings a day) [35]. Thus, the excess risk of frailty observed in the NHS may be of particular concern for a large fraction of the older US population with higher levels of SSBs intake.

So far, only 1 previous study has investigated the association between sweetened beverages and frailty. In a cohort of community-dwelling older people from Spain, participants consuming SSBs did not have an increased risk of frailty after 3 years of follow-up when compared with those who never consumed those beverages [36]. Besides its smaller sample size and a

**Table 3. RRs (95% CI) of frailty according to sweetened beverages consumption (serving/d), stratified by lifestyle factors, among 71,935 women aged ≥60 y in the NHS.**

| | Person years | Frailty cases | SSBs | P for interaction | ASBs | P for interaction | Total fruit juices | P for interaction | Orange juice | P for interaction | Other juices[a] | P for interaction |
|---|---|---|---|---|---|---|---|---|---|---|---|---|
| Age <70 | 598,149 | 2,815 | 1.15 (1.07, 1.23) | 0.01 | 1.07 (1.03, 1.11) | 0.27 | 0.93 (0.87, 0.99) | 0.35 | 0.82 (0.75, 0.90) | 0.03 | 1.10 (0.99, 1.21) | 0.53 |
| Age ≥70 | 442,472 | 8,744 | 1.09 (1.03, 1.16) | | 1.10 (1.06, 1.13) | | 0.97 (0.93, 1.01) | | 0.91 (0.86, 0.95) | | 1.11 (1.04, 1.19) | |
| BMI <25 kg/m² | 456,022 | 4,256[b] | 1.17 (1.08, 1.28) | 0.44 | 1.13 (1.07, 1.19) | 0.46 | 0.99 (0.93; 1.05) | 0.10 | 0.88 (0.82; 0.96) | 0.37 | 1.20 (1.09, 1.32) | 0.11 |
| BMI ≥25 kg/m² | 507,392 | 6,560 | 1.11 (1.05, 1.17) | | 1.09 (1.06, 1.12) | | 0.92 (0.88, 0.97) | | 0.86 (0.82, 0.91) | | 1.05 (0.97, 1.13) | |
| Low physical activity (<median) | 510,848 | 9,268 | 1.10 (1.04, 1.17) | 0.37 | 1.08 (1.05, 1.11) | 0.59 | 0.95 (0.91, 1.00) | 0.63 | 0.88 (0.83, 0.93) | 0.97 | 1.10 (1.02, 1.18) | 0.42 |
| High physical activity (≥median) | 527,805 | 2,291 | 1.16 (1.08, 1.25) | | 1.11 (1.06, 1.15) | | 0.95 (0.90, 1.01) | | 0.89 (0.83, 0.95) | | 1.10 (1.00, 1.20) | |
| Low AHEI level (<median) | 517,268 | 6,846 | 1.15 (1.09, 1.21) | 0.64 | 1.09 (1.06, 1.13) | 0.97 | 0.9 (0.93, 1.01) | 0.34 | 0.90 (0.85, 0.95) | 0.46 | 1.11 (1.03, 1.20) | 0.34 |
| High AHEI level (≥median) | 523,354 | 4,713 | 1.12 (1.02, 1.24) | | 1.09 (1.04, 1.13) | | 0.94 (0.89, 0.99) | | 0.87 (0.81, 0.94) | | 1.06 (0.97, 1.16) | |

Models were adjusted for age (years), calendar time (4-y intervals), BMI (<25.0, 25.0–29.9, ≥30.0 kg/m²), smoking status (never, past, and current 1–14, 15–24, and ≥25 cigarettes/day), alcohol intake (0, 1.0–4.9, 5.0–14.9, or ≥15.0 g/d), energy intake (quintiles of kcal/d), physical activity (quintiles), medication use (aspirin, postmenopausal hormone therapy, diuretics, β-blockers, calcium channel blockers, ACE inhibitors, other blood pressure medication, statins and other cholesterol-lowering drugs, insulin, and oral hypoglycemic medication), AHEI (quartiles), cancer, heart disease, and diabetes, except for the stratification variable. All beverages were mutually adjusted for each other.

[a] This group includes apple juice or cider, grapefruit juice, prune juice, and non-specified fruit juices.

[b] The number of events is different because of missing values for BMI.

95% CI, 95% confidence interval; ACE, angiotensin converting enzyme; AHEI, Alternate Healthy Eating Index; ASB, artificially sweetened beverage; BMI, body mass index; NHS, Nurses' Health Study; RR, relative risk; SSB, sugar-sweetened beverage.

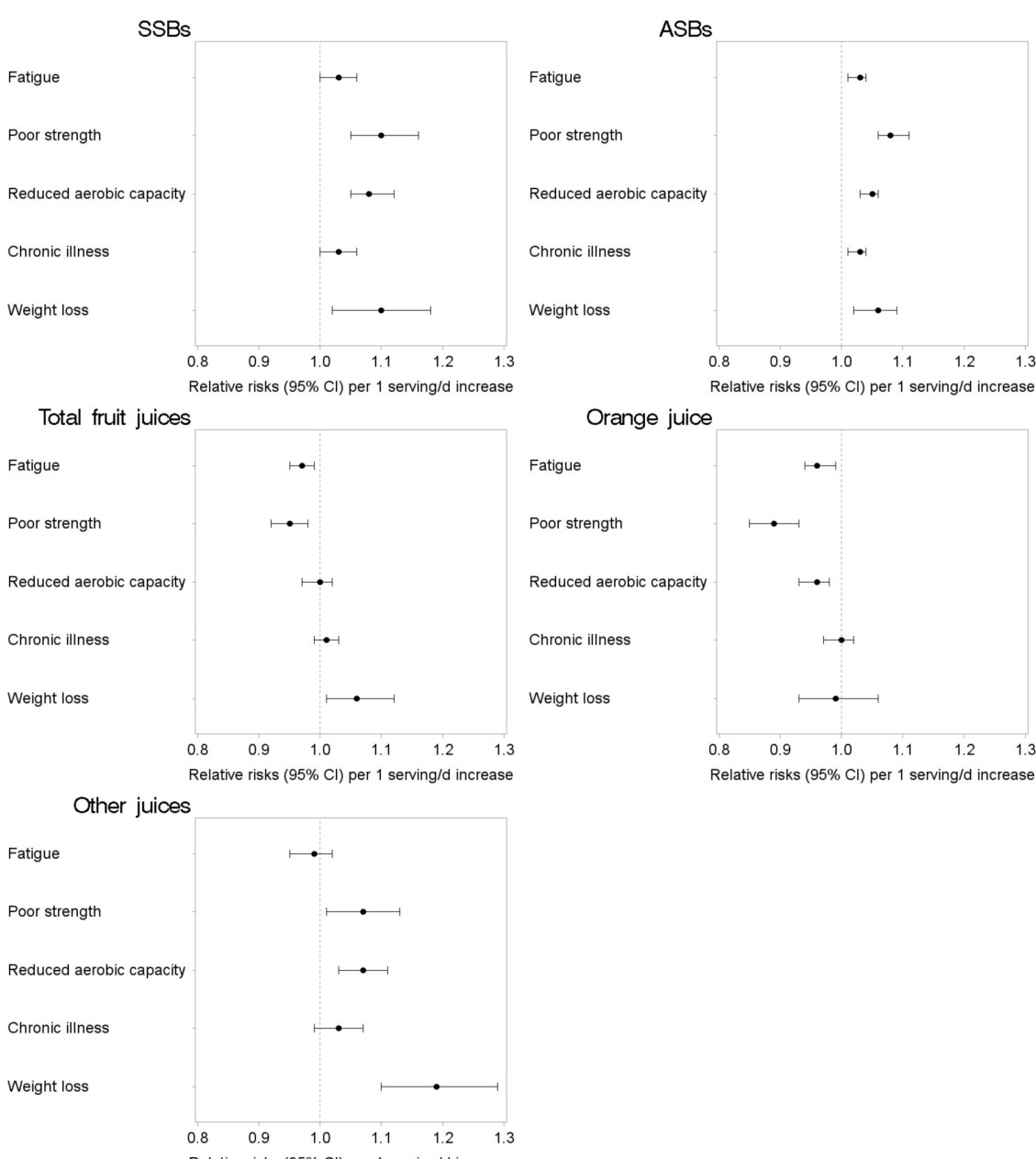

**Fig 1. RRs (95% CI) of frailty components according to sweetened beverages consumption (serving/d) among women aged ≥60 y in the NHS.** Adjusted for age (years), calendar time (4-y intervals), BMI (<25.0, 25.0–29.9, ≥30.0 kg/m²), smoking status (never, past, and current 1–14, 15–24, and ≥25 cigarettes/day), alcohol intake (0, 1.0–4.9, 5.0–14.9, or ≥15.0 g/d), energy intake (quintiles of kcal/d), physical activity (quintiles), medication use (aspirin, postmenopausal hormone therapy, diuretics, β-blockers, calcium channel blockers, ACE inhibitors, other blood pressure medication, statins and other cholesterol-lowering drugs, insulin, oral hypoglycemic medication), AHEI (quartiles), cancer, heart disease, and diabetes. All beverages were mutually adjusted for each other. The group "other juices" includes apple juice or cider, grapefruit juice, prune juice, and non-specified fruit juices. 95% CI, 95% confidence interval; ACE, angiotensin converting enzyme; AHEI, Alternate Healthy Eating Index; ASB, artificially sweetened beverage; BMI, body mass index; NHS, Nurses' Health Study; RR, relative risk; SSB, sugar-sweetened beverage.

short duration of follow-up, another plausible explanation for their results was the very low intake of SSBs observed. Moreover, in the Spanish study, frailty was defined using the Fried criteria, so their results might not be directly comparable with ours.

There is consistent evidence that liquid sources of carbohydrates are associated with less satiety than solid sources and that their intake is not compensated by reduced intake of other foods, so total daily energy intake is increased [13]. This is 1 of the mechanisms that may partially explain the association between SSBs and obesity, diabetes, and cardiovascular disease [16,37]. Frailty may also result from other biological pathways that contribute to the association between SSBs and those diseases. For example, added sugar in SSBs may lead to inflammation, impaired glucose, and lipid metabolism [14,15], leading to the occurrence of several chronic diseases and possibly increasing the risk of frailty. These mechanisms also impair muscle glucose handling and intracellular energy production and reduce protein synthesis, which leads to sarcopenia and less efficient muscle contraction [38–39]. Furthermore, high-fructose corn syrup used in SSBs produces a significant dose–response increase in uric acid concentrations [40], which has been associated with frailty incidence [41]. Our results also suggest that the association between sweetened beverages and frailty was not entirely mediated or due to obesity and other diseases since main analyses were adjusted for BMI, and the sensitivity analyses excluding participants with cardiovascular disease, diabetes, cancer, or overweight still showed a significant direct association.

While other types of fruit juices were not related to a lower risk of frailty, orange juice was inversely associated with risk. Many antioxidant nutrients and bioactive substances (including vitamins, carotenoids, flavonoids, and polyphenols) are found in juices and especially in orange juice. These compounds may limit oxidative stress and inflammation, which are core mechanisms of the decline in muscle function and strength in old people as well as of frailty [42–44]. Flavonoids and the carotenoid beta-cryptoxanthin may also lower the risk of cognitive decline [45]. Our results showed that orange juice was inversely associated with all the individual criteria of frailty, except for the illnesses criterion. Although the criteria mostly reflect physical frailty, cognitive impairment is closely related to physical frailty and might also play a role in the development of the individual criteria used to define frailty in this study [2]. Therefore, the potential beneficial effect of orange juice observed might be partly attributed to an improvement in cognitive status. However, our results need to be confirmed in further studies before public health recommendations can be made.

Similar to our results, other studies have found positive associations between ASBs and several outcomes including mortality, diabetes, and cardiovascular diseases [17,46,47]. Little is known about possible biological mechanisms that could explain these associations. It has been suggested that the potential adverse effects of ASB may be caused by a detrimental effect on gut microbiota that, in turn, may have a negative effect on glucose tolerance [48]. On the other hand, the authors suggested that misclassification and reverse causation could account for the results found [17,46]. In our study, the results for ASBs consumption and frailty held among different subgroups of participants, with healthy and unhealthy lifestyle behaviors.

Frailty is an important outcome because it is the consequence of alterations in many physiological systems. Frailty and pre-frail status as defined by the FRAIL scale has shown to be a significant predictor of disability among older adults [49]. Although reverting frailty development is challenging for many patients, at an early stage, frailty might be reversible and is therefore a valuable tool to identify those at risk for further adverse health effects [50]. Some previous research has shown that several diet-related factors (e.g., Mediterranean diet, fruits, and vegetables) that lower the risk of frailty [9,51–52] also lower the risk of disability [53,54]. Thus, we could speculate that that sweetened beverages might also have a detrimental effect on disability.

Strengths of this study are the large sample size, the repeated diet measurements that allowed calculating cumulative average consumptions, and the use of updated information on covariates in a cohort with high rates of follow-up. However, several limitations need to be acknowledged. First, since dietary information was self-reported, measurement error and misclassification could occur; however, the FFQ used here has been extensively validated against diet records and biomarkers, showing good correlations [20], and the repeated measures reduced random error. Second, although we were able to adjust for many potential confounders, and sensitivity analysis among subgroups of healthy participants showed robustness of the results, some unmeasured and residual confounding cannot be ruled out. Reverse causation cannot be totally discarded; however, latency analyses showed similar associations to main analyses. Third, although studying the risk of frailty among only female nurses helped to increase internal validity, the observed associations might not apply to other populations. Fourth, frailty is a dynamic condition and therefore, potentially reversible. However, as well as other chronic conditions, once it occurs, it is unlikely to reverse. Finally, performance-based measures were not available in this large cohort of older women. Due to the use of self-reported information, misclassification of frailty might have occurred. Our results should be confirmed in studies using other frailty definitions that include more objective measurements [2, 4].

In conclusion, we found that habitual consumption of ≥1 serving/d of SSBs, as well as ASBs, was associated with a higher risk of frailty. By contrast, consumption of orange juice was associated with lower risk of frailty. Whether ASBs consumption has a detrimental effect on frailty or is a spurious finding is unclear as this was not a prior hypothesis, and plausible mechanisms have not been established. Further studies of both SSBs and ASBs in relation to frailty would be valuable. Considering the high intake of SSBs in the US population and its many adverse health effects, possibly including frailty, older adults should be advised to limit their SSBs intake.

## Supporting information

**S1 Table. Relative risks (95% confidence interval) of frailty according to categories of the most recent information of sweetened beverages consumption before the onset of frailty among 67,739 women.**
(DOCX)

**S2 Table. Relative risks (95% confidence interval) of frailty according to categories of sweetened beverages consumption among 57,760 women with 0 frailty criteria at baseline.**
(DOCX)

**S3 Table. Relative risks (95% confidence interval) of frailty according to categories of sweetened beverages consumption among 82,430 women aged ≥60 y in the Nurses' Health Study, including diet before baseline in 1980, 1984, and 1986.**
(DOCX)

**S4 Table. Relative risks (95% confidence interval) for latency analysis of frailty according to sweetened beverage intake (serving/d) among women aged ≥60 y in the Nurses' Health Study.**
(DOCX)

**S5 Table. Relative risks (95% confidence interval) of frailty according to categories of sweetened beverages consumption among 60,402 women aged ≥60 y without heart disease, diabetes, or cancer in the Nurses' Health Study.**
(DOCX)

**S6 Table. Relative risks (95% confidence interval) of frailty with the weight loss criteria defined as a 10% weight reduction in 2 years according to categories of sweetened beverages consumption among 72,180 women aged ≥60 y in the Nurses' Health Study.**
(DOCX)

**S1 STROBE Checklist.**
(DOC)

## Author Contributions

**Formal analysis:** Ellen A. Struijk, Esther Lopez-Garcia.

**Methodology:** Ellen A. Struijk, Fernando Rodríguez-Artalejo, Teresa T. Fung, Walter C. Willett, Frank B. Hu, Esther Lopez-Garcia.

**Writing – original draft:** Ellen A. Struijk, Esther Lopez-Garcia.

**Writing – review & editing:** Fernando Rodríguez-Artalejo, Teresa T. Fung, Walter C. Willett, Frank B. Hu, Esther Lopez-Garcia.

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
