## [Editor Report · Decision Letter 0]

19 May 2020

Dear Dr Struijk, 

Thank you for submitting your manuscript entitled "Sweetened beverages and risk of frailty among older women in the Nurses´ Health Study" for consideration by PLOS Medicine.

Your manuscript has now been evaluated by the PLOS Medicine editorial staff as well as by an academic editor with relevant expertise and I am writing to let you know that we would like to send your submission out for external peer review.

Kind regards,

Thomas J McBride, PhD,

PLOS Medicine

---

## [Decision Letter · Decision Letter 1]

6 Jul 2020

Dear Dr. Struijk,

Thank you very much for submitting your manuscript "Sweetened beverages and risk of frailty among older women in the Nurses´ Health Study" (PMEDICINE-D-20-02037R1) for consideration at PLOS Medicine. 

Your paper was evaluated by a senior editor and discussed among all the editors here; it was also sent to independent reviewers, including a statistical reviewer. The reviews are appended at the bottom of this email and any accompanying reviewer attachments can be seen via the link below:

[LINK]

In light of these reviews, I am afraid that we will not be able to accept the manuscript for publication in the journal in its current form, but we would like to consider a revised version that addresses the reviewers' and editors' comments. Obviously we cannot make any decision about publication until we have seen the revised manuscript and your response, and we plan to seek re-review by one or more of the reviewers. 

We expect to receive your revised manuscript by Jul 27 2020 11:59PM. Please email us (plosmedicine@plos.org) if you have any questions or concerns.

We look forward to receiving your revised manuscript. 

Sincerely,

Emma Veitch, PhD

PLOS Medicine

On behalf of Clare Stone, PhD, Acting Chief Editor,

PLOS Medicine

plosmedicine.org

*Please revise your title according to PLOS Medicine's style - as well as setting out the study question this should also summarise the study design/framework used, after a colon (eg, ": prospective cohort). 

*At this stage, we ask that you include a short, non-technical Author Summary of your research to make findings accessible to a wide audience that includes both scientists and non-scientists. The Author Summary should immediately follow the Abstract in your revised manuscript. This text is subject to editorial change and should be distinct from the scientific abstract. Please see our author guidelines for more information: https://journals.plos.org/plosmedicine/s/revising-your-manuscript#loc-author-summary

*The abstract should have some minor changes to structure, please use the headings Background, Methods and Findings, Conclusions (nb, "Methods and Findings" is a single subsection). 

*In the last sentence of the Abstract Methods and Findings section, please include a brief note about any key limitation(s) of the study's methodology.

*Did your study have a prospective protocol or analysis plan? Please state this (either way) early in the Methods section.

*It's encouraging the paper has been reported using the STROBE guideline - however we'd suggest also appending the completed STROBE checklist as a supporting information file alongside the submitted revised paper - checklist can be downloaded at https://www.equator-network.org/reporting-guidelines/strobe/. When completing the checklist, please use section and paragraph numbers, rather than page numbers.

Comments from the reviewers:

Reviewer #1: In this manuscript, authors investigate the association of SSB, ASB and fruit juice intake and risk of frailty in women 60 years and older, using data from the Nurses' Health Study. They report a statistically significant increased risk of frailty with high consumption of SSB, ASB and fruit juice other than orange juice, and a lower risk of frailty with high consumption of orange juice. The manuscript is well written, and easy to read. There are number of strengths to this analysis: authors use a well-established sizable cohort with good quality data; prospective study design; long follow up; and repeated exposure measurement during follow up. They also conducted a comprehensive set of statistical analyses, including a number of sensitivity analyses. Nonetheless, physical activity variable notably misses from their multivariable models, despite being an important confounder (major limitation). They report from a stratified analysis by PA level (low/high), though this does not seem enough to rule out the confounding effect of PA. I suggest repeating all the models including METh/wk.

Table 3: it would be helpful to see p for interaction. Perhaps a footnote indicating a statistically significant interaction. 

Reviewer #2: Struijk et al. present a large prospective epidemiological study on sweetened beverages and risk for frailty. The study has indisputable strengths, including a very large sample size (n=almost 76,000 women), and a long follow-up (up to 22 years) between exposures (comprehensive exposure of most sweetened beverages) and ascertainment of frailty in older age. Very few studies on this topic have been conducted so far, therefore this report is important for the research field. 

It is generally acknowledged that frailty has many risk factors with small to moderate effect sizes, and given the aging of the population worldwide, the observed decreased risk for frailty with higher orange juice intake could lead to a strong reduction in the number of cases.

In sum, the paper is well written overall although the way nutritional aspects are discussed should be improved. The authors would benefit from deciphering underlying mechanisms, instead of just mentioning obesity, diabetes, cardiovascular diseases or inflammation, and anti-oxidant properties of fruits to explain their results. A more complicated approach of the food matrix effect of fruit intakes compared with fruit juices intake would have been interesting. Moreover, a discussion about the possible reversibility of frailty would have reinforce the discussion section. 

More specific remarks: 

Abstract: 

The "dose-response manner" mentioned in the conclusion is not ascertained by the retained results of the abstract. Please remove.

Introduction:

Are the artificially sweetened beverages (ASBs) largely consume among USA residents? The prevalence of consumption is only described for sugar-sweetened beverages SSBs and fruit juices.

A mention of the association between higher fruits intake or between carotenoids levels and the lower risk of frailty among European older people should be considered (Garcia-Esquinas et al 2016 and Pilleron et al 2019).

Methods:

Regarding the exposure, could the cumulative consumption of SSBs, ASBs and fruit juices be a relevant analysis? Indeed, some people may be considered both as SSBs consumers, OR ASBs consumers, while they consumed both beverages. I wondered how this cumulative exposure could influence the results.

Regarding the identification of incident frail participants, one out of five criteria is weight loss. I'm surprised to observe that the threshold retained was a reduction of 5% of weight during the last 2 years. How participants lost to follow-up for a visit were classified? -the delay between 2 visits being higher than 2 years? Moreover, why the authors have not considered the GLIM criteria (Cederholm et al 2018) to approach undernutrition (the consensus of experts proposed a reduction of weight of 5% for less than 6 months or 10% for 6 months and over).

Regarding frailty as a whole, it's acknowledged that subjective criteria are used, but it's surprising not to considered physical activity (MET are available) and a more objective scale than the FRAIL one (which also need a clinical judgment form the practitioner which seems not available here). The authors should acknowledge that the scale is a proxy of a more subjective scale and a misclassification could occur. Finally, the concordance between the FRAIL scale and the Fried criteria has been described in a care setting, but not yet among community-dwellers. This is another limitation.

More importantly, how were considered participants with frailty and disability (in instrumental activities of daily living IADL): it may occur an overlap between both frailty and IADL disability, while some authors argued that frailty is not disability (and the scale used to identify frail people should also not identify IADL disabled participants, see Zamudio-Rodríguez A et al, Age Ageing 2020). This important point should be discussed in depths. 

Regarding the statistical approach, I identified 2 major limits: the first one is the potential reversibility of the frailty status. It's acknowledged that frailty might be transitory (even more when the duration of the follow-up is high), and then, the identification of frail people who are then censured at the first time they are identified frail could induce a bias. The second limit is the lack of consideration of the risk for death: however, when the duration of the study is so long, the proportion of older people who died during the follow-u is high, and in the present study, the exposure can also be associated with this specific risk of death. Therefore, the analysis is mainly appropriate among the survivors, and the competitive risk for death compared with frailty cannnot be dismissed. To limit this bias, an illness-death analysis should be performed. It would reinforce the results.

Results.

Table 1 provides the description of low, moderate or high consumers of specific beverages, while the statistical part provided data on 6 levels of consumption: how were these 6 different levels rearranged on 3 different levels of intake?

Table 1: is alcohol expressed as g/d? please specify.

The text relative to the description of Table 2 (and others) would benefit from details (for instance, 3 different RR are provided, but it's not easy to understand which exposure is linked to which RR).

The test of interaction (p value) between exposure, BMI … should be provided to ensure the relevance of such stratified analysis.

Discussion.

In addition to the limits already discussed above, the relevance of such results among women only should be addressed.

The discussion about the differences observed between orange juice or other fruits juices intake appears limited and only in the field of cognition. This should be discussed further, because of the public health relevance of such results. I suggest revising with deeper discussion on nutritional aspects. 

Overall, it remains minor issues that should be addressed to improve this paper which is already of high quality. 

Reviewer #3: The authors present an investigation of associations between SSBs, ASBs and fruit juices, and frailty in the NHS. This is a well written, well argued, coherent manuscript which generates useful results with potential value to public health policy makers. I recommend publication of his paper with only a couple of very minor suggestions for improvement below. 

Page 6, para 2: the FFQ is 'reasonably valid' - this is vague; what does reasonably valid mean?

Page 9, line 1: the number of participants included in the study belongs in the results, not the methods.

Reviewer #4: This is a well-conducted study on the association between sweetened beverages and risk of frailty among older women using the Nurses´ Health Study (NHS) cohort. However, there are a few issues needing attention.

1) Competing risk. All the relative risks (RR) in table 2 were approximated by hazard ratios from the Cox proportional hazards models. However, as the outcome is frailty rather than all cause mortality, there is an issue of competing risk (from death). During the 22 years follow-up, potentially many participants (n=?) died. Therefore competing risk analyses need to be performed to derive the true HR/RR.

2) Many baseline variables were adjusted in table 2 as shown in the Multivariable model a and b. However, models were not adjusted for baseline co-morbidities which is not adequate. Although authors argued that the frailty outcome consists of some of the conditions, the baseline co-morbidities are different and need to be adjusted. In sensitivity analyses (supplementary table 5) the authors made some effort to address the issue but it is a bit patchy not systematic. It would be good to have a multivariable model c to additionally adjust these co-morbidities at baseline.

3) Not clear whether SSBs, ASBs, orange juice and etc are mutually exclusive in the questionnaires. What if a person takes two more different types of drinks during the follow-ups? How did the authors adjust this potential overlap in the analyses?

4) Table 1, in the Dietary intake section, most variables appeared to be skewed with non-normal distributions therefore should be summarised as median and IQR rather than mean and SD.

[LINK]

---

## [Decision Letter · Decision Letter 2]

13 Oct 2020

Dear Dr. Struijk,

Thank you very much for re-submitting your manuscript "Sweetened beverages and risk of frailty among older women in the Nurses´ Health Study: a prospective cohort study" (PMEDICINE-D-20-02037R2) for review by PLOS Medicine.

I have discussed the paper with my colleagues and the academic editor and it was also seen again by the statistical reviewer. I am pleased to say that provided the remaining editorial and production issues are dealt with we are planning to accept the paper for publication in the journal.

[LINK]

We look forward to receiving the revised manuscript by Oct 20 2020 11:59PM. 

Sincerely,

Thomas McBride, PhD

Senior Editor 

PLOS Medicine

plosmedicine.org

Requests from Editors:

1- With apologies for recommending this in the previous revisions, please remove “prospective” from the subtitle.

2- Thank you for providing your STROBE checklist. Please replace the page numbers with paragraph numbers per section (e.g. "Methods, paragraph 1"), since the page numbers of the final published paper may be different from the page numbers in the current manuscript. Please also refer to the checklist (S1 Checklist) when you mention it in the Methods section, and add the checklist to the list of supplemental files at the end of the main text.

3- Thank you for noting that there was no formal prospectively written protocol and mentioning analyses that were added at the request of reviewers. Should you also note the combined analysis of SSB and ASB intakes, added in response to reviewer 2?

4- Please include the results of the sensitivity analysis for weight loss defined as 10% weight reduction (in response to reviewer 2) in the supplementary information (referenced from the main text).

5- Similarly, please note in the main text and include in supplementary information the assessment of frailty reversal that was also included in response to reviewer 2.

6- Abstract Methods, please note the setting of the Nurses Health Study, and provide some demographic information (eg, average age).

7- In the Abstract Conclusions, please address the study implications without overreaching what can be concluded from the data; the phrase "In this study, we observed ..." may be useful. The last sentence of the Abstract Conclusions could specify what types of further studies are necessary.

8- Author summary, point 2, perhaps, “... an increasing number of people are at risk of developing frailty.”

9- Author summary, point 5, “... while higher orange juice consumption was associated with a lower risk.”

10- Author summary, point 6, no need to limit this to the US population, SSB intake is high in many countries. (Similarly for the Discussion conclusions).

11- Methods could note that participants provided consent as part of enrollment.

12- Methods, Dietary Assessment section. The bracketed sections at the bottom of page 7 (“ [e.g., Coke, Pepsi, and other colas with sugar],”; “ [e.g., 7-Up],” ; “ [e.g., Hawaiian Punch, lemonade, and other noncarbonated fruit drinks] ”)and beginning of page 8 should be parentheticals.

13- Results, page 17, please include the p value for the comparison of highest tertile vs. lowest tertile of ASB and SSBs

14- Please start of the Discussion: "In this analysis of a large prospective cohort in the United States, we found ..."

Comments from Reviewers:

Reviewer #4: Many thanks authors for their great effort to improve the manuscript. I am satisfied with the response and the revision. All my concerns were comprehensively addressed. No further issues needing attention.

[LINK]

---

## [Editor Report · Decision Letter 3]

29 Oct 2020

Dear Dr. Struijk, 

On behalf of my colleagues and the academic editor, Dr. Carol Brayne, I am delighted to inform you that your manuscript entitled "Sweetened beverages and risk of frailty among older women in the Nurses´ Health Study: a cohort study" (PMEDICINE-D-20-02037R3) has been accepted for publication in PLOS Medicine. 

PRODUCTION PROCESS

Before publication you will see the copyedited word document (within 5 business days) and a PDF proof shortly after that. The copyeditor will be in touch shortly before sending you the copyedited Word document. We will make some revisions at copyediting stage to conform to our general style, and for clarification. When you receive this version you should check and revise it very carefully, including figures, tables, references, and supporting information, because corrections at the next stage (proofs) will be strictly limited to (1) errors in author names or affiliations, (2) errors of scientific fact that would cause misunderstandings to readers, and (3) printer's (introduced) errors. Please return the copyedited file within 2 business days in order to ensure timely delivery of the PDF proof. 

If you are likely to be away when either this document or the proof is sent, please ensure we have contact information of a second person, as we will need you to respond quickly at each point. Given the disruptions resulting from the ongoing COVID-19 pandemic, there may be delays in the production process. We apologise in advance for any inconvenience caused and will do our best to minimize impact as far as possible.

PRESS

PROFILE INFORMATION

Thank you again for submitting the manuscript to PLOS Medicine. We look forward to publishing it. 

Best wishes, 

Thomas McBride, PhD

Senior Editor 

PLOS Medicine

plosmedicine.org